# Moderation analysis of exchange rate, tourism and economic growth in Asia

Bosede Ngozi Adeleye[1,2]*, Jimoh Sina Ogede[3], Mustafa Raza Rabbani[4], Lukman Shehu Adam[5], Maria Mazhar[6]

1 Dept of Accountancy, Finance and Economics, University of Lincoln, Lincoln, United Kingdom, 2 Lincoln International Business School, University of Lincoln, Lincoln, United Kingdom, 3 Dept of Economics, Olabisi Onabanjo University, Ago-Iwoye, Nigeria, 4 Dept of Accounting and Finance, British University of Bahrain, Sar, Kingdom of Bahrain, 5 Dept of Economics and Development Studies, Kwara State University, Malete, Nigeria, 6 School of Economics, Quaid-i-Azam University, Islamabad, Pakistan

* NAdeleye@lincoln.ac.uk

**Data Availability Statement:** All relevant data are within the paper and its Supporting Information files.

**Funding:** The author(s) received no specific funding for this work.

## Abstract

This study brings novelty to the tourism literature by re-examining the role of exchange rate in the tourism-growth nexus. It differs from previous tourism-led growth narrative to probe whether tourism exerts a positive effect on economic growth when the exchange rate is accounted for. Using a moderation modelling framework, instrumental variables general method of moments (IV-GMM) and quantile regression techniques in addition to real per capita GDP, tourism receipts and exchange rate, the study engages data on 44 Asian countries from 2010 to 2019. Results from the IV-GMM show that: (1) tourism exerts a positive effect on growth; (2) exchange rate depreciation hampers growth; (3) the interaction effect is positive but statistically not significant; and (4) results from EAP and SA samples are mixed. For the most part, constructive evidence from the quantile regression techniques reveals that the impact of tourism and exchange is significant at lower quantiles of 0.25 and 0.50 while the interaction effect is negative and statistically significant only for the SA sample. These are new contributions to the literature and policy recommendations are discussed.

## 1. Introduction

The tourism and hospitality industry has experienced development and expansion making it one of the biggest and fastest-growing sectors [1]. Many countries and destinations have grown in popularity, resulting in an increase in the number of visitors and tourism receipts. The tourism sector has the potentials to make significant contributions to economic growth and development through a variety of channels. It is a "currency earning sector" that permits the use of human and physical capital stock to drive innovation and development. Simultaneously, the tourism sector is either directly or indirectly related to other sectors like transportation, accommodation, or retailing through trickledown effect [2]. It also influences spending, and expands trade and global competitiveness [3]. International tourism, in particular, is a source of foreign exchange generation which improves the balance of payment position

**Competing interests:** The authors have declared that no competing interests exist.

[4] and eases the acquirement of advance technologies and capital goods that can be used in other manufacturing processes [5, 6]. Furthermore, it plays an important role in stimulating investments in new infrastructure and enhancing competition thereby creating jobs and improving overall living standard [2].

Similarly, the exchange rate influences economic growth. In this paper, an improvement/increase in the exchange rate indicates the appreciation of a domestic currency against a foreign currency. It is a significant indicator of economic progress as it essentially mirrors the competitiveness between a domestic economy and the rest of world. The exchange rate reflects a standard exchange among purchasers and merchants of foreign currency in the foreign exchange market of a particular country. Particularly, non-oil trades, oil exporters, international tourist expenditures, and foreign remittances all drive inflow of foreign currency. According to Rapetti et al. [7] the growth effect of exchange rate specifically the real exchange rate (RER) is both growth-amplifying and growth-dwindling. The exchange rate can significantly affect a country's balance of payments position particularly if the country's reliance on imported goods is high. In these circumstances, a more competitive RER would aid in relieving foreign exchange bottlenecks that would otherwise stymie the development process.

The connection between tourism and the exchange rate is not far-fetched. International tourism receipts are significant sources of foreign exchange earnings and highly linked to the exchange rate. Changes in exchange rates greatly affect tourism demand in a destination as changes in the exchange rate will have an impact on the currency value of the country of origin. Any adjustments in the exchange rate will prompt an appreciation or depreciation of the tourist's currency, affecting transportation costs and the tourist's decisions to visit the country. Thus, the exchange rate has an impact on the number of tourists' visits as well as tourism receipts [8]. Less flexible exchange rates are supposed to advance global exchange and tourism by lessening vulnerability in worldwide transactions, wiping out exchange costs, and expanding market transparency. Furthermore, the exchanges rate mimics the relative price differential (as it affects global economic environment, purchasing power and overall wealth of tourists), which tourists have insufficient information about since they make travel arrangements in their own currency in advance before leaving their country. In this way, low-uncertainty exchange rate regimes could promote international tourism flows [9] that in turn speed up the development process through foreign direct investment and globalization [10].

Tourism as a commodity is very susceptible to exchange rate shocks which affects tourists' inclination to visit a foreign country. We, therefore, hypothesize that changes in the exchange rate will influence the impact of tourism on economic growth. To the best of our knowledge, this is the first study to empirically test this hypothesis. That is, does the exchange rate tilt the tourism-growth dynamics? To probe the discourse, an unbalanced panel data on 44 Asian economies from 2010 to 2019 comprising tourism receipts, per capita GDP (proxy for economic growth), official exchange rate and a set of control variables is used. To ensure the robustness of the results, a blend of econometrics techniques is deployed. To control for possible endogeneity of the tourism variable, the instrumental variable technique nested within the generalised method of moments (IV-GMM) is used [11–13]. Lastly, the quantile estimator [14–16] is used in the event that the dependent variable has a non-normal distribution. This empirical approach makes the study novel and holistic in ensuring a critical examination of its core arguments. The rest of the paper is structured as follows: section 2 discusses the literature; section 3 outlines the data and empirical model; section 4 discusses the results, and section 5 concludes.

## 2. Literature review

Tourism activities are considered as one of the most important sources of economic growth and foreign exchange earnings around the globe [2, 6, 17]. The literature on tourism development and its impact on exchange rate and economic growth has increased exponentially in the last three decades [18, 19]. The studies on tourism and growth nexus have proliferated mainly due to the fact that international tourism has grown over the years despite some ephemeral shocks [20]. The tourism growth literature mainly focuses on the causal relationship between tourism and economic growth [19, 21–23] whereas, tourism and exchange rate literature focus mainly on exchange rate volatility and tourist flows [24–26]. We divide our literature review into two parts; the first part consists of available literature on tourism and economic growth whereas, the second part consists of tourism and exchange rate.

### 2.1 Tourism and economic growth

This section discusses the literature on tourism economics focusing on economic growth and tourism nexus. From a theoretical perspective, Lanza and Pigliaru [27] were among the first to document the tourism-growth nexus. They find that countries with high tourism sectors experienced high economic growth. They developed a Lucas type-two sector model where tourism is taken as one of the sectors which depends on the endowments of natural resources such that countries with abundant natural resources have high growth potential and achieve a faster rate of growth. Perles-Ribes et al. [28] studied the tourism and economic growth nexus using autoregressive distributed lag (ARDL) and Toda-Yamamoto model for the period 1957 to 2014 taking into consideration the economic crises. Their findings revealed a bi-directional relationship between economic growth and tourism development. There are many studies proposing the hypothesis that growth of tourism in the country is directly linked to economic prosperity [29]. The study reports that there is bidirectional causality between tourism and economic growth. Fuinhas et al. [22] report that in the long run, high frequency of tourist arrivals in the country leads to positive economic growth. In another study, Naseem [30] concludes that in the long run, tourism receipts, number of tourist arrivals, and total expenditure have a strong positive relationship with economic growth. The study empirically examined the data from Saudi Arabia and validated the popular hypothesis that tourism leads to economic growth in the country. Similar findings were obtained by [31–35], where they concluded that tourism has a positive impact on the economic growth of the country. The study by Sahni et al. [36] used a quantile regression approach and concluded that tourism growth has a more pronounced effect on economic growth below the threshold and above the threshold. The study further concluded that countries with lower economic growth have more benefits from tourism development. The study by Selvanathan et al. [37], applied ARDL, vector error correction model (VECM) and panel frameworks and concluded that in the long run tourism development positively contributes to growth. Tourism development is the significant predictor of the economic growth and financial development at frequency rather than the low frequency [38]. On the contrary, Croes et al. [39], revealed that tourism development has a very short term effect on economic development and a negative and indirect link to human development. Similar findings were obtained by Kyara et al. [23] where it was revealed that there is a unidirectional causality relationship between tourism development and economic growth.

### 2.2 Tourism and exchange rate

The effects of exchange rate on tourism development can differ across the country, territory and within the tourism jurisdiction [38]. The real and nominal appreciation of the currency leads to a negative impact on the tourism development in the country [40]. Exchange rate has

asymmetric impact on tourism on tourism development in developing countries such as, India, Bangladesh, Pakistan and Nepal in the short run [41]. Boskurt et al. [42] applied dynamic common correlated effects (DCCE) approach in their study on demand and exchange rate shocks on tourism development and concluded that effects of the exchange rate shocks are temporary on the tourism development. To examine the response of tourism demand to exchange rate fluctuation in South Korea, Chi [43] used ARDL model and concluded that tourists are sensitive to the appreciation of the Korean Won, whereas they are insensitive to its depreciation. The findings of the study imply that foreign visitors in Korea are loss averse and with increase or decrease in the exchange rate volatility tend to affect the tourism demand in an asymmetric manner. Dogru et al. [44] used ARDL approach to examine the trade balance and exchange rate taking evidence from tourism development. The study concluded that depreciation and appreciation of the US Dollar affects the bilateral tourism with Canada, Mexico, and the United Kingdom (UK). The study further concluded that in the long-run the appreciation of the US dollar negatively affects the tourism trade balance with Canada and the UK while it does not affect the tourism development with Mexico in the long-run. A study by Belloumi [45], examined tourism receipts and exchange rate nexus in Tunisia and concluded that there is a cointegrating relationship between tourism and economic growth. An increase in foreign direct investment (FDI) and appreciation of the exchange rate contracts the tourism demand of the country while in the long-run the depreciation of domestic currency and decrease in FDI inflow results in more tourist inflow [41]. Similar findings were obtained by [46] and [47] where they revealed that reduction in FDI inflow and depreciation of foreign exchange rate results in positive tourism development.

## 2.3 Tourism, exchange rate and economic growth

There are few studies that investigated the nexus of exchange rate, tourism development and economic growth [23, 48, 49]. Primayesa et al. [50] probed the dynamic relationship among real exchange rate, economic growth and tourism development in Indonesia using variance decomposition and impulse response function approach. The study revealed that in explaining the tourism shock in Indonesia, the real exchange rate is less important than the economic growth. The study further concluded that the shock of economic growth and real exchange rate has a positive effect on tourism activity in the short- and long-term. Harvey et al. [25] applied bounds testing approach to cointegration and error-correction modelling to examine whether tourism development and exchange rate promote the economic growth in Brunei Darussalam, Indonesia, Malaysia, and the Philippines. The study revealed the Philippines is the only country that has the positive long-run and short-run impact from the tourism industry and exchange rate.

## 3. Data and methodology

This study uses data on nine variables sourced from World Development Indicators (WDI) for 44 countries located in East Asia and the Pacific (EAP) and South Asia (SA) from 2010 to 2019. Availability of sufficient data on the variables of interest–per capita GDP, tourism receipts, and official exchange rate—justify the inclusion of a country in the sample and to explore the heterogeneity of the sample countries, we disaggregate the full sample into EAP with 36 countries and SA having 8 countries. The countries are **East Asia and the Pacific (36):** American Samoa, Australia, Brunei Darussalam, Cambodia, China, Fiji, French Polynesia, Guam, Hong Kong SAR, China, Indonesia, Japan, Kiribati, Korea, Dem. People's Rep., Korea, Rep., Lao PDR, Macao SAR, China, Malaysia, Marshall Islands, Micronesia, Fed. States, Mongolia, Myanmar, Nauru, New Caledonia, New Zealand, Northern Mariana Islands, Palau,

Papua New Guinea, Philippines, Samoa, Singapore, Solomon Islands, Thailand, Timor-Leste, Tonga, Vanuatu, Vietnam. **South Asia (8):** Afghanistan, Bangladesh, Bhutan, India, Maldives, Nepal, Pakistan, Sri Lanka.

## 3.1 Dependent variable

Real GDP per capita is the proxy for economic growth. Studies on tourism-growth nexus have widely used it [51–53] likewise, those on exchange rate-growth relationship [54, 55].

## 3.2 Main explanatory variables

From World Development Indicators, International tourism, receipts (% of total exports) is defined as: expenditures by inbound visitors including payments to foreign carriers for international transport. In other words, this composite variable captures the spendings of inbound tourists to Asia and the Pacific, among others. In line with the literature [56–60], tourism receipts which is the first main explanatory variable is proxied by tourism receipts in current US dollars. Existing literature have found a positive relationship between different dimensions of tourism and economic growth [61–65]. The second key explanatory variable is exchange rate [42, 66–68]. The exchange rate captures the competitiveness of a country in the international market [69–73]. Lastly, to address the study questions, an interaction term of tourism receipts with exchange rate *(TRPT*XR)* is included to determine if exchange rate moderates the impact of tourism on growth.

## 3.3 Control variables

The set of control variables align with those used in growth models: mobile phone subscription [5, 74, 75], individuals using the Internet [76, 77], labour force participation [78] (Niebel, 2018), foreign direct investment net inflows [79], domestic credit to the private sector [80–83] and services trade [84, 85]. We expect positive coefficients in line with existing literature. Table 1 details the variables used.

## 3.4 Empirical model

We specify two baseline linear models that expresses economic growth as a function of tourism receipts, exchange rate and a set of control variables which satisfies the first objective:

$$\ln PC_{it} = \alpha_0 + \alpha_1 \ln TRPT_{it} + \alpha_2 \mathbf{Z}'_{it} + \varphi_t + u_{it} \qquad [1]$$

**Table 1. Variables description, and expectations.**

| Variable | Description | Signs |
|---|---|---|
| *PC* | GDP per capita (constant 2010 US$) | N/A |
| *TRPT* | International tourism, receipts (current US$) | + |
| *XR* | Official exchange rate (LCU per US$, period average) | +/- |
| *DC* | Domestic credit to the private sector (% of GDP) | + |
| *LAB* | Labor force participation rate, total (% of total population ages 15–64) (modeled ILO estimate) | + |
| *FDI* | Foreign direct investment, net inflows (BoP, current US$) | + |
| *MOB* | Mobile cellular subscriptions | + |
| *NET* | Individuals using the Internet (% of population) | + |
| *TRS* | Trade in services (% of GDP) | + |

Source: Authors' Compilation from World Bank [86] World Development Indicators (WDI)

$$\ln PC_{it} = \gamma_0 + \gamma_1 XR_{it} + \alpha_3 \mathbf{K}'_{it} + \delta_t + e_{it} \tag{2}$$

Where, $\ln PC_{it}$ = natural logarithm of per capita GDP; $\ln TRPT_{it}$ = natural logarithm of tourism receipts; $XR_{it}$ = official exchange rate; $\mathbf{Z}'_{it}$ and $\mathbf{K}'_{it}$ = vector of control variables in natural logarithms; $\alpha_i$, $\gamma_i$ = parameters to be estimated; $\varphi_t$, $\delta_t$ = year dummies (which controls for common shocks such as the global financial crises of 2007–2009), and $u_{it}$, $e_{it}$ = general error term. To satisfy the second objective, we add an interaction term ($TRPT^*XR$) to Eq [1] and the model becomes:

$$\ln PC_{it} = \eta_0 + \eta_1 \ln TRPT_{it} + \eta_2 XR_{it} + \eta_3 (\ln TRPT_{it} * XR_{it}) + \eta_4 \mathbf{R}'_{it} + \omega_t + v_{it} \tag{3}$$

Where, $\mathbf{R}'_{it}$ = vector of control variables in natural logarithms; $\eta_i$ = parameters to be estimated; $\omega_t$ = year dummies (which controls for common shocks such as the global financial crises of 2007–2009), and $v_{it}$ = general error term. From Eq [3], $\eta_3$ provides two information. First, the sign of the coefficient indicates if exchange rate exerts a significant moderation effect on economic growth. That is, whether the interaction of both variables intensifies or hinders growth. Secondly, the magnitude of the coefficient may sustain or sway the impact of tourism on growth which is derived as:

$$\frac{\partial \ln PC}{\partial \ln TRPT} = \eta_1 + \eta_3 XR \tag{4}$$

## 3.5 Estimation techniques and strategy

Specifically, our econometric strategy consists of a three-step procedure. First, we examine linear impact of tourism on economic growth. Next, we estimate the linear effect of exchange rate on economic growth. Lastly, we perform the moderation analysis to show the interaction effect on economic growth. We engage these analyses using two techniques: the instrumental variables-two-step generalised method of moments (IV-GMM) techniques and the quantile estimator [14–16]. Specifically, the IV-GMM technique is used to correct for cross-sectional dependence, endogeneity, autocorrelation and heteroscedasticity in the data [11, 87]. It uniquely deploys the *ivreg2* routine in Stata version 16 developed by Baum, Schaffer, and Stillman [12, 13]. The routine performs several variants of single-equation linear regression models including the generalized method of moments (GMM). Hence, the GMM variant which implements the two-step feasible GMM estimation (that is, *gmm2s* option) is adopted to ensure that our results are devoid of endogeneity, heteroscedasticity and autocorrelation [12]. On the other hand, the quantile regression is deployed to examine the potentially differential effects of tourism and exchange rate at different levels of growth. The quantile regression model is a defined solution to minimize the equation for the $\theta$th regression quantile, $0 < \theta < 1$ and expressed thus:

$$\min_{b \in R^K} \left[ \sum_{t \in (t:y_t > x_t b)} \theta |y_t - x_t b| + \sum_{t \in (t:y_t < x_t b)} 1 - \theta |y_t - x_t b| \right] \tag{5}$$

Where, $y_t$ is the dependent variable and $x_t$ is a $k$ x 1 vector of explanatory variables.

## 4. Results and discussions

### 4.1 Summary statistics and correlation analysis

The upper panel of Table 2 contains the correlation matrix's results, illustrating the relationship between the regressors and the outcome variables. Our findings indicate a negative correlation between per capita GDP and official exchange rate, implying that rising income will

**Table 2. Pairwise correlation analysis and summary statistics.**

| Variables | PC | TRPT | XR | DC | LAB | FDI | MOB | NET | TRS |
|---|---|---|---|---|---|---|---|---|---|
| PC | 1.000 | | | | | | | | |
| TRPT | 0.482*** | 1.000 | | | | | | | |
| XR | -0.192*** | 0.169*** | 1.000 | | | | | | |
| DC | 0.647*** | 0.678*** | 0.101*** | 1.000 | | | | | |
| LAB | 0.277*** | 0.399*** | 0.288*** | 0.48*** | 1.000 | | | | |
| FDI | 0.434*** | 0.853*** | 0.201*** | 0.535*** | 0.242*** | 1.000 | | | |
| MOB | 0.005 | 0.722*** | 0.269*** | 0.271*** | 0.034 | 0.770*** | 1.000 | | |
| NET | 0.790*** | 0.575*** | -0.009 | 0.689*** | 0.300*** | 0.431*** | 0.111* | 1.000 | |
| TRS | 0.228*** | -0.191*** | -0.199*** | 0.145** | 0.117** | -0.293*** | -0.620*** | 0.170*** | 1.000 |
| | | | | *Summary Statistics* | | | | | |
| Observations | 402 | 350 | 390 | 316 | 370 | 387 | 347 | 327 | 357 |
| Mean | 12398.47 | 9.08E+09 | 1295.019 | 71.422 | 69.06 | 1.54E+10 | 92490468 | 38.86 | 30.269 |
| Std. Dev. | 17111.58 | 1.42E+10 | 4011.636 | 51.718 | 10.528 | 4.24E+10 | 2.54E+08 | 28.493 | 26.864 |
| Minimum | 528.737 | 600000 | 0.966 | 3.201 | 46.38 | -4.16E+09 | 6200 | 0 | 2.338 |
| Maximum | 71992.15 | 6.52E+10 | 23050.24 | 233.211 | 87.976 | 2.91E+11 | 1.65E+09 | 96.023 | 145.643 |

Note

*** p<0.01

** p<0.05

* p<0.1

ln = Natural logarithm; PC = per capita GDP; TRPT = tourism receipts; XR = Official exchange rate; DC = Domestic credit to the private sector; LAB = labour force participation rate; FDI = foreign direct investment; MOB = mobile phone subscriptions; NET = individuals using the Internet; TRS = trade in services; 9.08E +09 = 9,080,000,000.00

Source: Authors' Computations

decrease the exchange rates in Asia. Likewise, individuals use the internet and the official exchange rate. Trade in services is negatively associated with tourism receipts, official exchange rate, FDI, and MOB. These findings suggest that increasing individuals using the internet and trade in services will impact the official exchange rate, tourism receipts, FDI, and MOB.

The lower panel of Table 2 indicates the summary statistics for the variables from 2010 to 2019. The average of per capita GDP, tourism receipts, official exchange rate, domestic credit to the private sector, labour force participation, foreign direct investment, mobile phone subscriptions, internet users, and trade in services are 12398.47, 9080000, 1295.02, 71.42, 69.06, 1540000, 92490468, 38.86, and 30.269, respectively, from the entire sample. At the same time, the standard deviation provides information on the deviation from sample averages.

## 4.2 IV-GMM results

Table 3 displays results for the instrumental variables-two-step generalised method of moments (IV-GMM). Across the Full, EAP, and SA samples, tourism receipts and exchange rate are instrumented with their first difference and level terms. Limiting to the variables of interest, the summary of the linear models from the full sample shows tourism receipts as a significant positive predictor of economic growth. The findings indicate that a percentage change leads to 0.88% rise in economic growth, on average, *ceteris paribus*. We argue that a well-structured tourist sector together with investments in modern infrastructure will boost growth supporting Tugcu [88], Alfaro [89], Calero and Turner [90], Cheng and Zhang [91], and Scarlett [92] all of which argue in favour of tourism-driven growth. The exchange rate shows a

**Table 3. IV-GMM results for the full and sub-samples (Dep Var: lnPC).**

| Variables | Full Sample | | | East Asia and the Pacific | | | South Asia | | |
|---|---|---|---|---|---|---|---|---|---|
| | Linear Models | | Moderation | Linear Models | | Moderation | Linear Models | | Moderation |
| | [1] | [2] | [3] | [4] | [5] | [6] | [7] | [8] | [9] |
| lnDC | -0.0622 | 0.173** | 0.0125 | -0.396** | 0.269** | -0.0618 | 0.306** | 0.277*** | 0.609*** |
| | (-0.488) | (2.538) | (0.123) | (-1.998) | (2.547) | (-0.295) | (2.639) | (5.266) | (3.409) |
| lnLAB | -2.274*** | -0.353 | -1.467** | -3.331*** | -0.130 | -1.677 | -1.839*** | -1.840*** | -4.569*** |
| | (-3.366) | (-1.183) | (-2.248) | (-2.907) | (-0.281) | (-1.545) | (-4.492) | (-6.617) | (-3.259) |
| lnFDI | 0.0525 | 0.304*** | 0.125* | 0.0413 | 0.266*** | 0.131* | 0.225*** | 0.239*** | 0.0738 |
| | (0.728) | (10.24) | (1.799) | (0.485) | (6.705) | (1.664) | (6.961) | (7.837) | (0.884) |
| lnMOB | -0.756*** | -0.342*** | -0.619*** | -0.722*** | -0.312*** | -0.510*** | -0.355*** | -0.334*** | -0.707*** |
| | (-7.802) | (-11.57) | (-6.590) | (-7.285) | (-7.671) | (-4.746) | (-5.730) | (-11.52) | (-4.964) |
| lnNET | 0.165 | 0.927*** | 0.366* | 0.268 | 0.894*** | 0.504** | 0.364* | 0.360*** | -0.544 |
| | (0.701) | (11.94) | (1.692) | (1.007) | (9.150) | (2.268) | (1.703) | (4.553) | (-1.497) |
| lnTRS | -0.828*** | -0.232*** | -0.641*** | -0.671*** | -0.276*** | -0.474*** | 0.0977 | 0.340*** | -0.0608 |
| | (-4.737) | (-3.760) | (-3.890) | (-3.861) | (-3.429) | (-3.102) | (0.803) | (4.566) | (-0.472) |
| lnTRPT | 0.881*** | | 0.616*** | 0.961*** | | 0.520** | 0.0218 | | 1.300** |
| | (5.599) | | (3.481) | (5.203) | | (2.166) | (0.302) | | (2.267) |
| XR | | -5.40e-05*** | -0.000813 | | -6.29e-05*** | -0.00104 | | 0.00719*** | 0.285** |
| | | (-6.267) | (-1.488) | | (-6.580) | (-1.459) | | (6.936) | (2.140) |
| lnTRPT*XR | | | 3.39e-05 | | | 4.34e-05 | | | -0.0131** |
| | | | (1.424) | | | (1.404) | | | (-2.102) |
| Constant | 12.63*** | 6.025*** | 9.647*** | 15.58*** | 5.278*** | 10.05** | 13.65*** | 12.39*** | 8.972*** |
| | (4.235) | (4.397) | (3.503) | (3.381) | (2.734) | (2.486) | (6.751) | (10.25) | (4.826) |
| Year Dummies | Yes | Yes | Yes | Yes | Yes | Yes | Yes | Yes | Yes |
| Observations | 221 | 223 | 221 | 166 | 168 | 166 | 55 | 55 | 55 |
| R-squared | 0.540 | 0.825 | 0.708 | 0.420 | 0.786 | 0.702 | 0.942 | 0.977 | 0.946 |
| Hansen-J | 0.2021 | 0.3182 | 0.2638 | 0.2033 | 0.2766 | 0.3874 | 0.0850 | 0.3014 | 0.1242 |
| F-Statistic | 69.66*** | 92.90*** | 81.72*** | 42.67*** | 63.14*** | 59.51*** | 235.4*** | 187.9*** | 337.6*** |

Note

*** $p<0.01$

** $p<0.05$

* $p<0.1$

*t*-statistics in (); -5.40e-05 = 0.0000540; ln = Natural logarithm; PC = real per capita GDP; TRPT = tourism receipts; XR = Official exchange rate; DC = Domestic credit to the private sector; LAB = labour force participation rate; FDI = foreign direct investment; MOB = mobile phone subscriptions; NET = individuals using the Internet; TRS = trade in services.

Source: Authors' Computations

significant negative effect on growth. According to the findings, a percentage-point change in the exchange rate results in a 0.00005% drop in economic growth. The reason for this is not far-fetched. Exchange rate fluctuations influence potential travellers' decisions to alter their destination or shorten their vacation resulting in revenue loss for economies. This may result in adjustments to visitors' travel plans while in a particular nation [93]. These findings corroborate those of Lin, Liu, and Song [94], Meo et al. [95], Sharma and Pal [96], Chi [43], and Seraj and Coskuner [97]. For EAP countries, tourism increases economic growth by 0.62%, on average, *ceteris paribus*. On the other hand, the coefficient of the exchange rate is negative and significant at 1 per cent, which supports the argument of Vieira et al. [98] and Seraj and Coskuner [97]. These studies contend that local currency appreciation will decrease the

spending power of international tourists with consequent decline on tourism demand and economic growth. In South Asia, the effect of tourism on growth is positive but statistically not significant but exchange rate significantly boosts growth by 0.007%, on average, *ceteris paribus*. This finding contradicts Seraj and Coskuner [97] and suggests that currency appreciation is growth-enhancing. For the moderation models, columns [3, 6, 9] reveal that the interaction effect is positive but statistically not different from zero for the full and EAP samples while it decreases growth in South Asia which contradicts Sharma, Vashishat, and Rishad [99]. In other words, the conditional effect of tourism on growth reduces when exchange rate appreciates in South Asia.

On the reliability of the instruments used to validate the robustness of our estimations, we controlled for identification and exclusion restrictions which are indispensable for robust GMM estimations [12, 13]. Having used the IV-GMM estimation in *ivreg2*, the appropriate test of overidentifying restrictions and testing the validity of instruments used is the Hansen J statistic: the GMM criterion function. From the lower panel of Table 3, the *p*-value of the Hansen-J statistic across the six models ranges between 0.085 and 0.3874 which is clearly above 0.05. Hence, it fails to reject the null hypothesis of instruments validity indicating that the instruments used are valid and robust to our analysis.

## 4.3 Quantile regression results

Table 4 presents the quantile regression results across the 25th, 50th, and 75th quantiles of economic growth. The topmost panel displays the full sample results where tourism significantly improves growth at the 25[th] and 50[th] quantiles by 0.23% and 0.12%, respectively. Noticeably, the positive effect of tourism receipts *declines* along the distribution. On the other hand, exchange rate appreciation shows a reducing effect on growth at the 25[th] and 50[th] quantiles by -0.000051% and -0.000059%, respectively. This reducing effect is *larger* at the 50[th] quantile indicating that economic growth vulnerable to exchange rate fluctuations. Following our findings, we hypothesise that variations in the official exchange rate affects tourist purchasing decisions and economic growth in the long-run [100]. On the interaction effect, we find no significant impact on growth corroborating the results shown in Table 3.

The results of East Asia and the Pacific displayed in the middle panel indicate that tourism significantly increases growth at the 25[th] and 50[th] quantiles by 0.44% and 0.31%, respectively. A *reducing* positive effect is observed similar to that of the full sample. Also, exchange rate appreciation shows a reducing effect on growth at the 25[th] and 50[th] quantiles by -0.000061% and -0.000067%, respectively. Similar to the full sample, this reducing effect is *larger* at the 50[th] quantile and we find no significant interaction effect on growth. From the lowest panel, the results from South Asia indicate that tourism significantly increases growth at the 50[th] and 7[th] quantiles by 0.17% and 0.19%, respectively. An *increasing* positive effect is observed contrary to the full and EAP samples. Likewise, exchange rate appreciation increases economic growth across all the quantiles, though with a declining trend from 0.0087% to 0.0075%. Contrary to the full and EAP samples, a significant negative interaction effect is observed across the quantiles supporting the results shown in Table 3.

## 5. Conclusion and policy recommendation

This current study highlights the role of exchange rate in influencing the effect of tourism on economic growth in Asia. To the best of our knowledge, this is the first study that critically evaluates the influence of exchange rate on the tourism-growth nexus. That is, it gauges the nonlinear effect of tourism on economic growth when the exchange rate is accounted for. This position differs from other tourism-growth studies [22, 27–30, 101, 102] that investigated the

**Table 4. Quantile regression results for the full and sub-samples (Dep Var: lnPC).**

| Variables | Tourism | | | Exchange Rate | | | Moderation | | |
|---|---|---|---|---|---|---|---|---|---|
| | q25 | q50 | q75 | q25 | q50 | q75 | q25 | q50 | q75 |
| lnDC | 0.0477 | 0.0221 | 0.0943 | 0.173** | 0.114 | 0.0153 | 0.165** | -0.0134 | -0.0188 |
| | (0.412) | (0.190) | (0.536) | (2.046) | (1.416) | (0.139) | (2.064) | (-0.107) | (-0.107) |
| lnLAB | -0.760** | -0.459 | -1.262* | -0.319 | 0.00301 | -0.220 | -0.556** | -0.192 | -0.591 |
| | (-2.171) | (-1.319) | (-1.664) | (-1.075) | (0.00826) | (-0.347) | (-2.241) | (-0.519) | (-0.810) |
| lnFDI | 0.238*** | 0.318*** | 0.328*** | 0.334*** | 0.431*** | 0.348*** | 0.291*** | 0.325*** | 0.365*** |
| | (4.882) | (6.695) | (6.215) | (9.828) | (10.74) | (6.507) | (7.458) | (7.327) | (6.620) |
| lnMOB | -0.427*** | -0.516*** | -0.485*** | -0.339*** | -0.462*** | -0.337*** | -0.391*** | -0.497*** | -0.451*** |
| | (-6.376) | (-10.05) | (-6.684) | (-7.790) | (-10.62) | (-6.594) | (-7.180) | (-11.22) | (-5.639) |
| lnNET | 0.573*** | 0.664*** | 0.723*** | 0.654*** | 0.700*** | 0.892*** | 0.566*** | 0.648*** | 0.760*** |
| | (7.944) | (8.329) | (5.921) | (6.102) | (8.599) | (7.670) | (7.822) | (7.501) | (7.107) |
| lnTRS | -0.282** | -0.433*** | -0.267*** | -0.0572 | -0.338*** | -0.229*** | -0.132 | -0.394*** | -0.356*** |
| | (-2.394) | (-4.200) | (-4.123) | (-0.505) | (-4.280) | (-3.484) | (-1.073) | (-4.378) | (-4.420) |
| lnTRPT | 0.232*** | 0.217*** | 0.146 | | | | 0.125** | 0.210** | 0.120 |
| | (2.864) | (2.745) | (1.502) | | | | (2.033) | (2.487) | (1.157) |
| XR | | | | -5.05e-05*** | -5.87e-05*** | -3.31e-05 | -0.00109 | -0.000779 | -0.000582 |
| | | | | (-8.351) | (-3.964) | (-1.367) | (-1.153) | (-1.076) | (-0.724) |
| lnTRPT*XR | | | | | | | 4.63e-05 | 3.22e-05 | 2.33e-05 |
| | | | | | | | (1.116) | (1.012) | (0.661) |
| Constant | 7.004*** | 6.358*** | 10.01*** | 5.255*** | 5.115*** | 5.603** | 5.879*** | 5.020*** | 7.053** |
| | (4.592) | (3.981) | (2.914) | (3.977) | (3.488) | (2.212) | (5.577) | (3.100) | (2.152) |
| Replications | 100 | 100 | 100 | 100 | 100 | 100 | 100 | 100 | 100 |
| Observations | 252 | 252 | 252 | 254 | 254 | 254 | 252 | 252 | 252 |
| **East Asia and the Pacific** | | | | | | | | | |
| lnDC | -0.141 | -0.133 | -0.139 | 0.348** | 0.118 | -0.157 | 0.100 | -0.241 | -0.396 |
| | (-0.742) | (-0.550) | (-0.408) | (2.346) | (0.566) | (-0.869) | (0.415) | (-0.665) | (-0.996) |
| lnLAB | -0.366 | -0.166 | -1.087 | 0.507 | -0.204 | -0.0982 | 0.484 | -0.133 | 0.252 |
| | (-0.457) | (-0.293) | (-0.727) | (0.971) | (-0.286) | (-0.115) | (0.742) | (-0.231) | (0.217) |
| lnFDI | 0.302*** | 0.302*** | 0.254*** | 0.315*** | 0.371*** | 0.278*** | 0.273*** | 0.323*** | 0.234** |
| | (5.606) | (5.035) | (3.076) | (5.756) | (5.956) | (3.612) | (5.195) | (4.887) | (2.362) |
| lnMOB | -0.672*** | -0.560*** | -0.457*** | -0.374*** | -0.391*** | -0.187* | -0.485*** | -0.516*** | -0.319** |
| | (-7.079) | (-7.263) | (-4.088) | (-5.270) | (-6.537) | (-1.913) | (-5.175) | (-4.555) | (-2.502) |
| lnNET | 0.739*** | 0.747*** | 0.867*** | 0.759*** | 0.751*** | 0.873*** | 0.689*** | 0.724*** | 0.866*** |
| | (6.112) | (6.415) | (4.239) | (5.634) | (5.762) | (6.624) | (4.908) | (6.229) | (3.985) |
| lnTRS | -0.420*** | -0.353*** | -0.226** | -0.171 | -0.394*** | -0.197** | -0.239 | -0.324*** | -0.232** |
| | (-2.731) | (-3.454) | (-2.292) | (-0.852) | (-2.853) | (-2.062) | (-1.440) | (-2.654) | (-2.473) |
| lnTRPT | 0.443*** | 0.309** | 0.201 | | | | 0.235* | 0.271 | 0.184 |
| | (3.718) | (2.268) | (1.248) | | | | (1.666) | (1.446) | (1.241) |
| XR | | | | -6.10e-05*** | -6.71e-05*** | -0.000105*** | 0.000187 | -0.000440 | -0.000514 |
| | | | | (-5.745) | (-4.297) | (-3.940) | (0.152) | (-0.468) | (-0.761) |
| lnTRPT*XR | | | | | | | -1.04e-05 | 1.73e-05 | 1.89e-05 |
| | | | | | | | (-0.193) | (0.419) | (0.638) |
| Constant | 3.872 | 4.268* | 9.657 | 1.952 | 6.208** | 5.142 | 1.174 | 4.356* | 3.820 |
| | (1.203) | (1.663) | (1.487) | (0.888) | (2.065) | (1.588) | (0.399) | (1.713) | (0.827) |
| Replications | 100 | 100 | 100 | 100 | 100 | 100 | 100 | 100 | 100 |
| Observations | 189 | 189 | 189 | 191 | 191 | 191 | 189 | 189 | 189 |
| **South Asia** | | | | | | | | | |
| lnDC | 0.139 | 0.176** | 0.287* | 0.261** | 0.282*** | 0.310*** | 0.292*** | 0.346*** | 0.399*** |
| | (1.231) | (2.228) | (1.831) | (2.222) | (4.452) | (4.627) | (3.537) | (4.660) | (5.023) |

*(Continued)*

**Table 4.** (Continued)

| Variables | Tourism | | | Exchange Rate | | | Moderation | | |
|---|---|---|---|---|---|---|---|---|---|
| | **q25** | **q50** | **q75** | **q25** | **q50** | **q75** | **q25** | **q50** | **q75** |
| lnLAB | -1.417*** | -1.373*** | -1.692* | -1.678** | -1.960*** | -2.009*** | -2.223*** | -2.455*** | -2.780*** |
| | (-2.946) | (-2.784) | (-1.896) | (-2.495) | (-5.120) | (-5.200) | (-5.850) | (-6.130) | (-5.764) |
| lnFDI | 0.188*** | 0.208*** | 0.189*** | 0.265*** | 0.229*** | 0.212*** | 0.197*** | 0.187*** | 0.159*** |
| | (2.789) | (3.899) | (2.850) | (4.201) | (5.451) | (5.544) | (3.555) | (5.383) | (4.432) |
| lnMOB | -0.391*** | -0.434*** | -0.501*** | -0.334*** | -0.331*** | -0.311*** | -0.431*** | -0.409*** | -0.396*** |
| | (-5.331) | (-6.543) | (-5.871) | (-6.201) | (-9.191) | (-9.125) | (-9.044) | (-10.06) | (-9.568) |
| lnNET | 0.464*** | 0.342*** | 0.224* | 0.287** | 0.310*** | 0.315*** | 0.274** | 0.240*** | 0.185*** |
| | (4.405) | (3.113) | (1.679) | (2.561) | (4.145) | (4.559) | (2.381) | (3.548) | (2.750) |
| lnTRS | -0.125 | -0.148 | -0.242 | 0.455*** | 0.398*** | 0.416*** | 0.0866 | 0.212* | 0.289*** |
| | (-0.767) | (-1.325) | (-1.541) | (4.005) | (4.781) | (4.504) | (0.649) | (1.953) | (2.859) |
| lnTRPT | 0.155 | 0.167*** | 0.188*** | | | | 0.328*** | 0.316*** | 0.292*** |
| | (1.514) | (2.719) | (3.634) | | | | (2.730) | (3.958) | (2.765) |
| XR | | | | 0.00870*** | 0.00783*** | 0.00750*** | 0.0666** | 0.0734*** | 0.0656** |
| | | | | (4.577) | (6.032) | (4.875) | (2.465) | (3.540) | (2.482) |
| lnTRPT*XR | | | | | | | -0.00286** | -0.00313*** | -0.00274** |
| | | | | | | | (-2.223) | (-3.214) | (-2.235) |
| Constant | 11.38*** | 11.59*** | 14.33*** | 11.05*** | 13.05*** | 13.19*** | 10.55*** | 11.12*** | 13.13*** |
| | (5.353) | (5.611) | (3.980) | (3.657) | (7.331) | (8.085) | (5.096) | (6.929) | (7.786) |
| Replications | Yes | Yes | Yes | Yes | Yes | Yes | Yes | Yes | Yes |
| Observations | 63 | 63 | 63 | 63 | 63 | 63 | 63 | 63 | 63 |

Note

*** p<0.01

** p<0.05

* p<0.1

I-statistics in (); ln = Natural logarithm; PC = per capita GDP; TRPT = tourism receipts; XR = Official exchange rate; DC = Domestic credit to the private sector; LAB = labour force participation rate; FDI = foreign direct investment; MOB = mobile phone subscriptions; NET = individuals using the Internet; TRS = trade in services.

Source: Authors' Computations

direct and linear effect of tourism on economic growth but aligns with Adeleye et al. [103] who examined a similar nexus on Sri Lanka. For the most part, these studies affirm that tourism exerts a direct and positive effect on economic growth. However, we expand the frontiers of knowledge having recognized that the exchange rate is an important macroeconomic policy instruments for promoting sustainable economic growth and encouraging tourism flows as it serves as an essential factor influencing the decision of tourists regarding tourism destinations. To this end, this paper examines the moderating effect of exchange rate and tourism receipts on economic growth in Asia from 2010 to 2019. From the full sample, findings from IV-GMM and quantile regressions techniques revealed that tourism significantly boosts economic growth, and the exchange rate indicates a negative effect. Deductively, we conclude that tourism is growth-enhancing which supports the tourism-led growth conjecture and that exchange rate appreciation is also growth-reducing. On the interaction effect, though the coefficient is positive but statistically insignificant it suggests that currency appreciation may possess inherent potentials in sustaining the positive effect of tourism on economic growth. Results from the East Asia and the Pacific and South Asia are diverse.

Based on the findings, the following recommendations are made for the government and stakeholders in Asia: (1) Provide a sound and efficient financial system which does not only provide adequate funding for promoting the tourism sector but also ensure easy accessibility to aid foreign tourist's transaction. (2) Initiate investment incentive policies for the tourism sector which will reduce the operating cost, investment outlay and provide security for the investment of tourist investors. (3) Initiate a well-managed exchange rate system that supports tourism flows and economic growth. For further studies and subject to data availability, the role of government regulation, real exchange rate and competitiveness in relation to the tourism-growth dynamics may be undertaken.

## Supporting information

**S1 Data.**
(XLSX)

## Author Contributions

**Conceptualization:** Bosede Ngozi Adeleye.

**Data curation:** Bosede Ngozi Adeleye.

**Formal analysis:** Bosede Ngozi Adeleye.

**Investigation:** Bosede Ngozi Adeleye, Jimoh Sina Ogede.

**Writing – original draft:** Bosede Ngozi Adeleye.

**Writing – review & editing:** Mustafa Raza Rabbani, Lukman Shehu Adam, Maria Mazhar.

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
