## [Decision Letter · Decision Letter 0]

30 Aug 2022

PONE-D-22-18174Moderation analysis of exchange rate, tourism and economic growth in AsiaPLOS ONE

Dear Dr. Adeleye,

Thank you for submitting your manuscript to PLOS ONE. After careful consideration, we feel that it has merit but does not fully meet PLOS ONE’s publication criteria as it currently stands. 

In view of the referees’ feedback and my own reading of your paper, we believe your paper is some way from being publishable. In particular, there are serious doubts about the underlying hypotheses on which the research is based, as well as about the methodology used.

While we consider the issues identified to be major in nature, we are willing to offer you a chance to rework the paper if you feel able to address them fully and robustly. Therefore, we invite you to submit a revised version of the manuscript that addresses the points raised during the review process.

We look forward to receiving your revised manuscript.

Kind regards,

J E. Trinidad Segovia

Section Editor

PLOS ONE

Journal Requirements:

Reviewers' comments:

Reviewer's Responses to Questions

**Comments to the Author**

1. Is the manuscript technically sound, and do the data support the conclusions?

Reviewer #1: Yes

Reviewer #2: Yes

Reviewer #3: Partly

2. Has the statistical analysis been performed appropriately and rigorously? 

Reviewer #1: Yes

Reviewer #2: N/A

Reviewer #3: I Don't Know

3. Have the authors made all data underlying the findings in their manuscript fully available?

Reviewer #1: Yes

Reviewer #2: Yes

Reviewer #3: No

4. Is the manuscript presented in an intelligible fashion and written in standard English?

Reviewer #1: No

Reviewer #2: Yes

Reviewer #3: Yes

5. Review Comments to the Author

Reviewer #1: Idea of the paper and statistical parts have been performed appropriately and rigorously. And there is no problem. But some formal parts have mistakes and errors. Some of them:

1- Abstract part should be improved.

2- Too many references. Can be reduced according to the journal index (Scopus, Sci, ssci etc.)

3- No need to citations in Data and Methodology parts. They must be in Literature Review part.

4- In text there are citation errors. For example more than 3 authors use et al. Some parts it is true but some parts wrong.

5- Some citations are missed in references especially in page 12.

6- Use "literature review" instead of "Review of literature".

7- At conclusion part comparisons with previous studies can be made.

Reviewer #2: The paper attempts to examine “moderation analysis of exchange rate, tourism and economic growth in Asia”. After reviewing, I find that this paper is interesting. The paper is readable ragarding the case of economic growth in Asian in the background of exchange rate, tourism and their interactive association.

See the attachment

Reviewer #3: The paper under consideration looks at the impact of tourism on GDP growth and the interactions of the impact with exchange rate. In my opinion this paper has important shortcomings that will prevent it from being published in the current form. My suggestion is rejection. The issues that lead to my decision are as follows:

1. The paper largely ignores the growth regression literature and certainly aims to be a part of it.

2. The value added generated in the tourism sector is in fact part of the overall value added of the economy. This is largely correlated with the international tourism. What sense does it have to regress GDP on a component of it? We can find out quite precisely what is the EXACT contribution of tourism to GDP and GDP growth.

3. The models are estimated by GMM. However, what are the instruments? The paper does not seems to use any sort of Arellano-Bond, Arellano Bower System-GMM. So the description is vague. And in particular, the panel System-GMM methods are mainly used to solve the endogeneity caused by the lagged dependent variable and not the inherent endogeneity of the economic problem posed here. So this part clearly needs clarification and justification. It does not suffice to write that „results are devoid of endogeneity, heteroscedasticity and autocorrelation.”

4. The measurement of both TRPT and GDP in USD should be discussed, i.e., the volume of tourist services may be positively related to depreciating exchange rate but its value in USD may not.

5. To what extend this is a different problem than analysis of any export-oriented sector? Why do we care?

6. How about controlling for real exchange rate, standard in BOP and competitiveness-related studies.

6. PLOS authors have the option to publish the peer review history of their article (what does this mean?). If published, this will include your full peer review and any attached files.

Reviewer #1: **Yes: **Volkan Dayan

Reviewer #2: No

Reviewer #3: No

---

## [Author Response · Author response to Decision Letter 0]

7 Oct 2022

Dear Editor,

I have uploaded a Word file containing point-by-point responses to the Reviewers' comments.

Thank you.

Dr. Ngozi Adeleye

---

## [Decision Letter · Decision Letter 1]

14 Nov 2022

PONE-D-22-18174R1Moderation analysis of exchange rate, tourism and economic growth in AsiaPLOS ONE

Dear Dr. Adeleye,

Thank you for submitting your manuscript to PLOS ONE. After careful consideration, we feel that it has merit but does not fully meet PLOS ONE’s publication criteria as it currently stands.  One of the reviewers consider that his/her comments have not been properly addressed. Please revise the previous comments and answer to the reviewer. Please submit your revised manuscript by Dec 29 2022 11:59PM. If you will need more time than this to complete your revisions, please reply to this message or contact the journal office at plosone@plos.org. Please include the following items when submitting your revised manuscript:A rebuttal letter that responds to each point raised by the academic editor and reviewer(s). You should upload this letter as a separate file labeled 'Response to Reviewers'.A marked-up copy of your manuscript that highlights changes made to the original version. You should upload this as a separate file labeled 'Revised Manuscript with Track Changes'.An unmarked version of your revised paper without tracked changes. You should upload this as a separate file labeled 'Manuscript'.

We look forward to receiving your revised manuscript.

Kind regards,

J E. Trinidad Segovia

Section Editor

PLOS ONE

Reviewers' comments:

Reviewer's Responses to Questions

**Comments to the Author**

1. If the authors have adequately addressed your comments raised in a previous round of review and you feel that this manuscript is now acceptable for publication, you may indicate that here to bypass the “Comments to the Author” section, enter your conflict of interest statement in the “Confidential to Editor” section, and submit your "Accept" recommendation.

Reviewer #1: All comments have been addressed

Reviewer #2: (No Response)

Reviewer #3: (No Response)

2. Is the manuscript technically sound, and do the data support the conclusions?

Reviewer #1: Yes

Reviewer #2: Yes

Reviewer #3: Partly

3. Has the statistical analysis been performed appropriately and rigorously? 

Reviewer #1: Yes

Reviewer #2: Yes

Reviewer #3: I Don't Know

4. Have the authors made all data underlying the findings in their manuscript fully available?

Reviewer #1: Yes

Reviewer #2: Yes

Reviewer #3: Yes

5. Is the manuscript presented in an intelligible fashion and written in standard English?

Reviewer #1: Yes

Reviewer #2: Yes

Reviewer #3: Yes

6. Review Comments to the Author

Reviewer #1: This study brings novelty to the tourism literature by re-examining the role of exchange rate in the tourism-growth nexus. It differs from previous tourism-led growth narrative to probe whether tourism exerts a positive effect on economic growth when the exchange rate is accounted for. All corrections are enough for publishing.

Reviewer #2: Dear Authors,

I feel satisfied with this version and your replies on my comments. Therefore, I have decided the acceptance for this paper.

Best,

Reviewer #3: I do not believe that the authors have taken my comments seriously. It is not enough to say: "we are using GMM" because it is a large class of models. What are the instruments? What are the moment conditions? Authors say they do not use GMM based on lagged values (Arellano Bond, Arellano Bover) but another approach. So what exactly is it?

7. PLOS authors have the option to publish the peer review history of their article (what does this mean?). If published, this will include your full peer review and any attached files.

Reviewer #1: **Yes: **Volkan Dayan

Reviewer #2: No

Reviewer #3: No

---

## [Author Response · Author response to Decision Letter 1]

15 Nov 2022

Dear Editor,

Uploaded a Word file containing the response to Reviewer's comments.

Thank you.

Dr. Ngozi ADELEYE

---

## [Decision Letter · Decision Letter 2]

24 Nov 2022

PONE-D-22-18174R2Moderation analysis of exchange rate, tourism and economic growth in AsiaPLOS ONE

Dear Dr. ADELEYE,

Thank you for submitting your manuscript to PLOS ONE. After careful consideration, we feel that it has merit but does not fully meet PLOS ONE’s publication criteria as it currently stands.

I have to congratulate the authors for their efforts and the reviewers for their valuable comments. This latest version clearly shows a very substantial improvement on the manuscript.

However, there is still a minor issue to be resolved. In particular, the reviewer is asking for some probes on the robustness of the results.

Therefore, we invite you to submit a revised version of the manuscript that addresses the points raised during the review process. Please submit your revised manuscript by Jan 08 2023 11:59PM. If you will need more time than this to complete your revisions, please reply to this message or contact the journal office at plosone@plos.org. Please include the following items when submitting your revised manuscript:A rebuttal letter that responds to each point raised by the academic editor and reviewer(s). You should upload this letter as a separate file labeled 'Response to Reviewers'.A marked-up copy of your manuscript that highlights changes made to the original version. You should upload this as a separate file labeled 'Revised Manuscript with Track Changes'.An unmarked version of your revised paper without tracked changes. You should upload this as a separate file labeled 'Manuscript'.If applicable, we recommend that you deposit your laboratory protocols in protocols.io to enhance the reproducibility of your results. Protocols.io assigns your protocol its own identifier (DOI) so that it can be cited independently in the future. For instructions see: https://journals.plos.org/plosone/s/submission-guidelines#loc-laboratory-protocols. Additionally, PLOS ONE offers an option for publishing peer-reviewed Lab Protocol articles, which describe protocols hosted on protocols.io. Read more information on sharing protocols at https://plos.org/protocols?utm_medium=editorial-email&utm_source=authorletters&utm_campaign=protocols.

We look forward to receiving your revised manuscript.

Kind regards,

J E. Trinidad Segovia

Section Editor

PLOS ONE

Journal Requirements:

Reviewers' comments:

Reviewer's Responses to Questions

**Comments to the Author**

1. If the authors have adequately addressed your comments raised in a previous round of review and you feel that this manuscript is now acceptable for publication, you may indicate that here to bypass the “Comments to the Author” section, enter your conflict of interest statement in the “Confidential to Editor” section, and submit your "Accept" recommendation.

Reviewer #3: All comments have been addressed

2. Is the manuscript technically sound, and do the data support the conclusions?

Reviewer #3: Yes

3. Has the statistical analysis been performed appropriately and rigorously? 

Reviewer #3: Yes

4. Have the authors made all data underlying the findings in their manuscript fully available?

Reviewer #3: Yes

5. Is the manuscript presented in an intelligible fashion and written in standard English?

Reviewer #3: Yes

6. Review Comments to the Author

Reviewer #3: Now with the paper specifying the actual method and the choice of instruments, the only thing that is lacking is providing test statistics to check for the validity of instruments, i.e., the J statistic or equivalent and the discussion on the validity of instruments. The authors seem to treat the IV-GMM technique as a solution to several econometric problems but they do not check whether the problems have been indeed solved.

7. PLOS authors have the option to publish the peer review history of their article (what does this mean?). If published, this will include your full peer review and any attached files.

Reviewer #3: No

---

## [Author Response · Author response to Decision Letter 2]

28 Nov 2022

Dear Editor,

I have uploaded a Word file containing responses to the comments of the Reviewer.

Thank you.

Dr. Ngozi ADELEYE

---

## [Decision Letter · Decision Letter 3]

19 Dec 2022

Moderation analysis of exchange rate, tourism and economic growth in Asia

PONE-D-22-18174R3

Dear Dr. ADELEYE,

We’re pleased to inform you that your manuscript has been judged scientifically suitable for publication and will be formally accepted for publication once it meets all outstanding technical requirements.

Kind regards,

J E. Trinidad Segovia

Section Editor

PLOS ONE

Additional Editor Comments (optional):

Reviewers' comments:

Reviewer's Responses to Questions

**Comments to the Author**

1. If the authors have adequately addressed your comments raised in a previous round of review and you feel that this manuscript is now acceptable for publication, you may indicate that here to bypass the “Comments to the Author” section, enter your conflict of interest statement in the “Confidential to Editor” section, and submit your "Accept" recommendation.

Reviewer #3: All comments have been addressed

2. Is the manuscript technically sound, and do the data support the conclusions?

Reviewer #3: Yes

3. Has the statistical analysis been performed appropriately and rigorously? 

Reviewer #3: Yes

4. Have the authors made all data underlying the findings in their manuscript fully available?

Reviewer #3: Yes

5. Is the manuscript presented in an intelligible fashion and written in standard English?

Reviewer #3: Yes

6. Review Comments to the Author

Reviewer #3: After the latest additions, I recommend the paper for publication. It seems that instruments pass the required tests.

7. PLOS authors have the option to publish the peer review history of their article (what does this mean?). If published, this will include your full peer review and any attached files.

Reviewer #3: No

---

## [Editor Report · Acceptance letter]

21 Dec 2022

PONE-D-22-18174R3 

Moderation analysis of exchange rate, tourism and economic growth in Asia 

Dear Dr. ADELEYE:

I'm pleased to inform you that your manuscript has been deemed suitable for publication in PLOS ONE. Congratulations! Your manuscript is now with our production department. 

Kind regards, 

on behalf of

Dr. J E. Trinidad Segovia 

Section Editor

PLOS ONE